# Intracellular Diversity of WNV within Circulating Avian Peripheral Blood Mononuclear Cells Reveals Host-Dependent Patterns of Polyinfection

**DOI:** 10.3390/pathogens12060767

**Published:** 2023-05-26

**Authors:** Dalit Talmi-Frank, Alex D. Byas, Reyes Murrieta, James Weger-Lucarelli, Claudia Rückert, Emily N. Gallichotte, Janna A. Yoshimoto, Chris Allen, Angela M. Bosco-Lauth, Barbara Graham, Todd A. Felix, Aaron C. Brault, Gregory D. Ebel

**Affiliations:** 1Center for Vector-Borne Infectious Diseases, Department of Microbiology, Immunology and Pathology, Colorado State University, Fort Collins, CO 80523, USA; 2Department of Biochemistry and Molecular Biology, College of Agriculture, Biotechnology & Natural Resources, University of Nevada, Reno, NV 89557, USA; 3United States Department of Agriculture, Animal and Plant Health Inspection Service, Wildlife Services, Lakewood, CO 80228, USA; 4Division of Vector-Borne Diseases, National Center for Emerging Zoonotic Infectious Diseases, Centers for Disease Control and Prevention, Fort Collins, CO 80521, USA

**Keywords:** West Nile virus, *Flavivirus*, RNA virus evolution

## Abstract

Arthropod-borne virus (arbovirus) populations exist as mutant swarms that are maintained between arthropods and vertebrates. West Nile virus (WNV) population dynamics are host-dependent. In American crows, purifying selection is weak and population diversity is high compared to American robins, which have 100- to 1000-fold lower viremia. WNV passed in robins leads to fitness gains, whereas that passed in crows does not. Therefore, we tested the hypothesis that high crow viremia allows for higher genetic diversity within individual avian peripheral blood mononuclear cells (PBMCs), reasoning that this could have produced the previously observed host-specific differences in genetic diversity and fitness. Specifically, we infected cells and birds with a molecularly barcoded WNV and sequenced viral RNA from single cells to quantify the number of WNV barcodes in each. Our results demonstrate that the richness of WNV populations within crows far exceeds that in robins. Similarly, rare WNV variants were maintained by crows more frequently than by robins. Our results suggest that increased viremia in crows relative to robins leads to the maintenance of defective genomes and less prevalent variants, presumably through complementation. Our findings further suggest that weaker purifying selection in highly susceptible crows is attributable to this higher viremia, polyinfections and complementation.

## 1. Introduction

The genetic diversity of RNA virus populations, including arthropod-borne viruses (arboviruses) within hosts, is well described [1,2,3,4,5,6,7,8,9] and contributes to virus fitness, pathogenesis and adaptation in response to changing environments [10,11,12,13,14,15,16,17]. West Nile virus (WNV, *Flaviviridae*, *Flavivirus*) has adapted to local mosquitoes and birds since its introduction into North America in order to maximize transmission and concomitant human disease [18,19]. WNV population structure is shaped by alternating replication in wild birds and mosquitoes [12,15,20,21,22,23,24]. Birds exhibit distinct disease phenotypes during infection and exert species-specific impacts on WNV genetic diversity and fitness [12,15,17,25,26,27]. American crows (*Corvus brachyrhynchos*) are highly susceptible to WNV-induced mortality and produce extremely high viremia during acute infection. Conversely, American robins (*Turdus migratorius*) are relatively resistant to severe disease and produce lower viremia [13,26]. WNV that has replicated exclusively in robins bears the signature of strong purifying selection, with few insertions, deletions and nonsynonymous mutations detected [15]. However, a subset of mutations arising during replication in robins reaches relatively high population frequency, resulting in fitness gains. In crows, however, WNV achieves high levels of population richness that includes abundant intrahost single nucleotide variants (iSNVs) and lethal intrahost length variants (iLVs) and fitness losses [15]. Therefore, distinct avian species that serve as enzootic hosts for WNV in nature may have remarkably distinct impacts on virus population structure, fitness and transmission [11,15,18,28,29,30,31].

The differences in viral load in robins compared to crows may significantly contribute to the distinct impacts of robins compared to crows in WNV populations. WNV viremia within crows, and viral loads within their tissues, typically vastly exceeds those present in robins [11,15,26,32,33]. We, therefore, hypothesized that the extent of polyinfection (infection of a single cell by multiple distinct WNV genomes) in individual crow cells vastly exceeds that which occurs within robin cells. As a result, the ability of natural selection to remove deleterious variants from the population could be reduced due to more frequent complementation of defective or low-fitness genomes with those that are of high or average fitness. In addition, higher-fitness WNV genotypes may be suppressed by the large number of low-fitness WNV genomes generated via error-prone replication. Notably, both of these phenomena have been documented in *in vitro* studies [15,27,31,34,35,36,37] and are consistent with our prior observations of *in vivo* replication of WNV in wild birds [15]. Nonetheless, whether polyinfection indeed occurs more frequently in crows compared to robins has not been addressed. Further, the impact of host viremia on the strength of selection has not been experimentally examined using ecologically relevant animals.

Therefore, in this study, we assessed the degree of polyinfection within individual cells in crows and robins *in* and *ex vivo* compared to cultured avian fibroblast cells using a newly developed barcoded version of WNV. Specifically, we examined the dynamics of infection in PBMCs, key targets of WNV replication in mammals and birds, to explore the relationship between multiplicity of infection (MOI)-dependent polyinfection and complementation [38,39,40,41,42,43]. While the specific cell tropism of WNV within avian PBMCs is not well understood, monocytes and dendritic cells are known targets in humans and horses [18,40,44]. Single cell analysis of PBMCs from birds infected with barcoded WNV revealed more viral genotypes simultaneously infecting crow cells than robins. Rare viral genotypes were also more likely to be maintained during crow infections while they were rapidly eliminated during the infection of robins. This finding suggests that the fitness of WNV variants may be host-dependent, and this dependence may be related to viremia and the frequency of polyinfection. These results suggest that natural selection may be weakened within highly susceptible host species due to high viremias and MOI, leading to frequent polyinfection of cells, increasing the likelihood of complementation. Our results also provide support for previous observations that document slower virus divergence in crows compared to robins and point to the significance of American crows for maintaining virus genetic diversity under natural conditions.

## 2. Materials and Methods

**FtC-3699 Infection of ex vivo PBMCs and DF1 cells.** FtC-3699 is a wildtype WNV isolated from *Culex* spp. mosquito pools collected in Fort Collins, Colorado, in 2012 and passaged twice on Vero cells (GenBank accession KR868734). PBMCs were separated from American crow and robin whole blood using a Histopaque-1077 (Sigma-Aldrich, Burlington, VT, USA) gradient as previously described [39]. PBMCs and chicken dermal fibroblasts (DF1) (ATCC^®^ CRL-12203™) were infected with WNV strain FtC-3699 at MOIs of 0.1, 1 or 10, and they were washed and supplemented with fresh RPMI medium containing 10% FBS as described [39]. Supernatants were harvested at the designated time points and stored at −80 °C for plaque assays and RNA extraction.

**Generation of a Molecularly Barcoded WNV (BC-WNV).** A barcoded WNV was generated as previously described using a previously described WNV infectious clone [45,46]. Briefly, a region was identified in the NS4b protein for the insertion of degenerate synonymous nucleotides at 11 consecutive third codon positions where any mutation would result in no alteration to the amino acid. PCR amplifications were performed with Q5 DNA polymerase (NEB, Ipswich, MA, USA) and assembly was performed using the HiFi DNA assembly master mix (NEB Ipswich, MA, USA). The digested assembly reaction was amplified via rolling circle amplification using the Repli-g mini kit (Qiagen, Germantown, MD, USA). The correct assembly was confirmed by assessing the banding pattern through restriction digestion, and the sequence was confirmed using Sanger sequencing. Infectious RNA was generated with *in vitro* transcription using the ARCA 2X T7 master mix (NEB, Ipswich, MA, USA) with subsequent transfection in 293T cells (ATCC^®^ CRL-3216™) using Lipofectamine 3000 (Thermo Fisher Scientific, Waltham, MA, USA). Virus was harvested and aliquots were stored at −80 °C.

**Animals.** Animal use was reviewed and approved by CSU Institutional Animal Care and Use Committee (15-5958; 18-8080A) according to the Guide for the Care and Use of Laboratory Animals of the National Institutes of Health. Wild-caught American crows and American robins from northern Colorado were housed in 20 m^2^ rooms and provided water and a mixture of dry dog food (crows), moistened dry cat food (robins), berries (robins) and mealworms (robins) ad libitum in addition to various enrichment activities. Birds were tested for antibodies against WNV using a plaque-reduction neutralization test according to standard practices. Only serologically negative animals were used in infection experiments. Prior to infection, groups of 2–3 birds were moved to 0.5 to 1 m^3^ cages within CSU biosafety level three (BSL3) facilities. After infection, birds were monitored several times daily for clinical symptoms. Jugular venipuncture was performed for blood collection. Birds were euthanized at 5 days post-infection.

**Bird Infection.** Birds were inoculated with BC-WNV via subcutaneous injection in the pectoral region with 10,000 plaque forming units (PFUs) in 100 µL medium containing 1% FBS, MEM+ 2 mM glutamine+ 10% FBS + 1% non-essential amino acids (NEAA), penicillin-streptomycin, sodium bicarbonate at a final concentration of 1.5–2.2 g/L and sodium pyruvate at a final concentration of 110 mg/L.

**DF1 Infection.** DF1 cells were infected with BC-WNV similarly to previous infection using FtC-3699 virus. DF1 cells were detached using TrypLE and trypsin was neutralized before centrifugation. After supernatant was aspirated, 4% paraformaldehyde (PFA) in PBS was added to the cell pellet and cells were used for flow cytometry and sorting.

**Blood Processing.** PBMCs were separated from peak infection whole blood via density gradient centrifugation using Histopaque-1077 (Sigma-Aldrich, Burlington, VT, USA) and utilized for flow cytometry and sorting, RNA extraction and library preparation.

**PBMC and DF1 Preparation for Flow Cytometry.** PBMC and DF1 cells were incubated on ice, pelleted and washed with a staining buffer containing 1xPBS, 1% RNAse-free BSA (Gemini Bio, West Sacramento, CA, USA) and 1:400 RNasin Plus (Promega, Madison, WI, U&SA) before storage at −80 °C. Before flow cytometry, cells were washed with PBS, then permeabilized using 1X PBS, 0.1% Triton X100 (Sigma-Aldrich, Burlington, VT, USA), 1% RNAse-free BSA and 1:400 of RNasin Plus. Cells were pelleted, washed and blocked with medium containing 2% FBS, and incubated with an anti-WNV capsid antibody (GTX-131947; diluted 1:1000) and a subsequent AlexaFluor 647-labeled secondary antibody (diluted 1:1000).

**Single Cell Sorting and RNA Extraction.** Cells were sorted through a 70-micron nozzle at the lowest speed possible using the BD FACSAria™ III sorter. We gated on and collected WNV-positive cells. We used a modified protocol previously used to sequence individual fixed and stained single brain radial glial cells [47]. Cells were sorted into 96-well plates containing lysis buffer with proteinase K solution in PKD buffer (1:16) (Qiagen, Germantown, MD, USA) and were stored at −80 °C. Samples were incubated for 1 h at 56 °C with the lid set to 66 °C in a thermos block for reverse crosslinking. Total RNA was extracted using the Mag-Bind Viral DNA/RNA 96 kit (Omega Bio-Tek, Norcross, GA, USA) on the KingFisher Flex Magnetic Particle Processor (Thermo Fisher Scientific, Waltham, MA, USA), according to manufacturer’s protocols, adjusted for small volumes.

**Screening for Positive Cells.** Only RNA derived from wells containing single WNV-positive cells, as determined by qRT-PCR for the 18S housekeeping gene and WNV copies, was used for library preparation, regardless of WNV cycle threshold value.

**Library Preparation of Single cell RNA.** Previously described methods were modified to adapt the Primer ID approach [48,49] to the Illumina MiSeq platform. Methods are provided below, and development is provided in Appendix A.

**cDNA Generation and Purification.** Thus, 5 µL of RNA was combined with 1 µL of 10 mM deoxynucleoside triphosphates (dNTPs), 1 µL of cDNA primer (ID_cDNAWNV_7374_Rev) (10 µM) and 3 µL of nuclease-free water. The 10 µL reaction volume was incubated for 5 min at 65 °C and then placed on ice for 2 min. A reverse transcription reaction mixture containing 1µL of Superscript III RT enzyme, 1µL of RNaseOut, 2 µL dithiothreitol (DTT), 4 µL of 25 mM MgCl_2_ and 2 µL of 10X SSIII buffer was added to the previous reaction volume (20 µL total reaction volume) and incubated for 50 min at 50 °C, followed by 5 min at 85 °C. Reactions were chilled on ice, spun down and incubated for 20 min at 37 °C after an addition of 1µL of RNaseH. cDNA was purified using Agencourt xp beads (Beckman Coulter, Indianapolis, IN, USA) at 1X concentration with elution into 12 µL of nuclease-free water.

**PCR Amplification Step 1.** Two steps were used for amplification of the target amplicon. First, 11 µL of cDNA was combined with 0.75 µL of 10 nM forward primer (R1_5′_WNV_for), 0.75 µL of 10 nM forward primer (5′_ID_Primer_Rev) and 12.5 µL of 2X KAPA HiFi HotStart mastermix (VWR, Radnor, PA, USA). PCR conditions were 95 °C for 3 min, 98 °C for 20 s, 72 °C for 45 s and 72 °C for 1 min with 35 cycles. Samples were purified using Agencourt XP beads (Beckman Coulter, Indianapolis, IN, USA) at 0.6X concentration and eluted in 20 µL of nuclease-free water.

**PCR Amplification Step 2.** The second round of PCR amplification served to add barcodes and adapters. Thus, 2 µL of purified PCR product from PCR amplification step 1 was combined with 9 µL of nuclease-free water, 0.75µL of 10 nM forward primer (Illumina index i5), 0.75 µL of 10 nM forward primer (Illumina index i7), 12.5 µL of 2X KAPA HiFi HotStart mastermix (VWR, Radnor, PA, USA) and the same PCR conditions were followed as in PCR step 1, repeated for 10 cycles. Samples were purified using AMPure XP beads at 0.6x concentration with elution into 22 µL of nuclease-free water.

Samples were pooled at a volume of 5 µL each and concentrated using AMPure XP beads at 1.5X concentration. Pooled samples were quantified using Qubit, and size distribution was verified via Tapestation. Additional size selection was performed using AMPure XP beads according to manufacturer’s guidelines. Libraries were quantified using the NEB library quantification kit.

**Library Pooling and Loading.** Libraries were pooled by volume and concentration was normalized to 2 nM. Libraries were denatured and a 15% PhiX control was spiked in. Samples were loaded at a 7 pM concentration using an Illumina MiSeq system.


**Bioinformatics pipeline.**


Next-generation sequencing data were processed similarly to our previous studies [50]. Briefly, we used a modified version of the “Primer_ID_Barcode Analysis” workflow https://bitbucket.org/murrieta/primer-id-with-barcoded-virus/src/master/ (accessed on 8 May 2023) that makes use of the template consensus sequence (TCS) pipeline [49], bbmap https://sourceforge.net/projects/bbmap/ (accessed on 8 May 2023) and custom perl scripts to process and analyze the PrimerID-generated barcode virus sequencing data. The above workflow was modified so that the TCS pipeline used the forward primer sequence and the cDNA primer sequence (Appendix A). The PrimerID_BarcodeGenerator shell script was updated to use “GATGCTGGGGACAAGTCACC” for the upstream flanking sequence and “TTTTGCCACTATGCCTACAT” for the downstream flanking sequence for barcode isolation when aligning to the WNV infectious clone (GenBank accession AF404756) barcode region (nucleotides 7188 to 7391).

**Data Reporting.** The data generated as part of this project are available from the authors. Unprocessed.fastq files containing amplicon reads are available from the authors upon request. Count matrix data that were used for the analyses in this work are available here: https://github.com/gebel67/WNV_avian_amplicon (accessed on 10 May 2023).

## 3. Results

### 3.1. Ex Vivo WNV Replication in PBMCs Demonstrates Host-Specific Accumulation of Non-Infectious Genomes

In preliminary studies using a wildtype WNV isolate FtC-3699, we assessed WNV replication in DF1 cells and *ex vivo* American robin and American crow PBMCs. DF1 cells demonstrated higher titers than *ex vivo* cultured crow and robin PBMCs (Appendix A). Interestingly, robin PBMCs had higher peak infectious titers and RNA loads than crow PBMCs (Appendix A). Higher MOI produced higher titers in all cell types (Appendix A). We calculated genome/PFU ratios of all three cell types sampled across the 5-day infection (Appendix A). WNV from crow PBMCs had significantly more genomes per infectious unit across the three different MOI compared to DF1 cells and robin PBMCs (2-way ANOVA, Tukey’s Multiple Comparisons). At MOI 0.01, robin PBMCs had the lowest genome/PFU. At MOI 10, the genome/PFU ratio from robin PBMCs and DF1 cells were similar, but both remained approximately 100-fold lower than crow PBMCs (Appendix A).

### 3.2. Establishing and Characterizing WNV Barcoded Virus (BC-WNV) Stock

To measure virus diversity within individual cells, we generated a molecularly barcoded WNV (BC-WNV), as previously described for ZIKV [46,51]. BC-WNV contains a segment in the NS4b region (7237–7269 bp, Figure 1A) with 11 consecutive synonymous degenerate nucleotides at every third codon position. BC-WNV replicates similarly to unmodified WNV infectious clone in Vero cells (MOI = 0.1) (Figure 1B). Analysis of genetic diversity in the barcode region of the stock virus using unique molecular identifiers (UMIs) indicated that a total of 4835 viral sequences representing 2236 total unique barcodes were present per 50 µL of stock solution. Three unique barcodes were sequenced over 100 times in the stock, comprising 14% of all viral sequences. Seven barcodes were detected 51–100 times (9%), 54 were detected 10–50 times (23%), 160 appeared 2–9 times (11%) and 2008 barcodes were detected 1 time and constituted 41% of the stock.

### 3.3. Viral Barcode RNA Abundance Varies between Cells and Different Wild Bird Hosts

We quantified WNV barcodes in PBMCs from infected crows and robins after four days of *in vivo* replication and from DF1 cells inoculated at MOIs of 1 and 10 (Figure 2 and Appendix A). Using flow cytometry, cells were screened for the presence of intracellular WNV viral protein (Appendix A), and viral RNA was extracted from infected cells. WNV RNA from ninety-four DF1 cells from each MOI along with 376 crow and 144 robin PBMCs were sequenced. qRT-PCR of WNV E gene copies within cells demonstrated that individual infected cells contained between 10^1^ and 10^5.5^ viral genomes/cell (Figure 3A). The mean genome load within PBMCs was 11,265 and 399 in crows and robins, respectively. DF1 cells had mean genome loads of 8530 and 5982 when infected at MOI of 1 and 10, respectively. There was a bimodal distribution of viral load in crow cells, with 39% of cells containing an average of 10^4.5^ genomes (“high” viral load) and 61% containing 10^2.5^ genomes (“low” viral load). Robin PBMCs had significantly fewer WNV copies compared to DF1 cells or crows, which contained the highest mean genome copies per cell. WNV barcode counts from individual cells (Figure 3B) supported the qRT-PCR-based observations on viral load within individual cells; however, there was a small number of cells without any barcodes detected despite detectable WNV vRNA via qRT-PCR.

Analysis of unique barcodes within cells revealed that crow PBMCs contained significantly higher barcode diversity and barcode complexity (Figure 3C,D) compared to robin PBMCs and DF1 cells, despite a slightly lower sequencing coverage depth compared to robins (Figure 3E). Viral load, as measured by total barcode counts, was significantly positively correlated with barcode diversity (Figure 3F).

We next examined the fate of specific barcodes during infection of DF1 cells, crows and robins. Barcodes detected at lower frequencies in the input BC-WNV were almost never found in DF1 cells or robin PMBCs (Figure 4A). In contrast, rare input barcodes were detected more frequently in crow PBMCs, sometimes in greater than 50% of cells. More common barcodes from the input tended to be maintained in DF1 cells and crow PBMCs compared to robins, where even common input barcodes were often not detected (Figure 4B). WNV within DF1 cells tended to contain a single dominant barcode sequence that rose to extremely high frequency (Figure 4B, diamonds within DF1 cell panel), a distinguishing feature of DF1 cells compared with bird PBMCs. Finally, we used t-Distributed Stochastic Neighbor Embedding (t-SNE) analysis to plot the data in two-dimensional space (Appendix A). Although we applied different perplexities to our data, cells did not cluster into distinct groups. Nevertheless, the crow cells consistently grouped closer to the input stock sample than other cell types, supporting our observations that barcode sequences, including those that are quite rare in the input stock virus population, are maintained during crow replication but not in robins or DF1 cells.

## 4. Discussion

Virus–host interactions clearly shape viral populations and evolution. We previously demonstrated that high WNV viremias in crows, coupled with error-prone virus replication, result in an extremely complex set of virus mutants that contain abundant nonviable virus genomes in addition to a wide array of low-frequency virus variants [15]. This contrasts with robins, in which lower viremia is associated with fewer genomes that contain lethal mutations and fewer overall variants. However, in robins, the detected variants tended to rise to a higher frequency. PBMCs are an important site for WNV replication in birds. We hypothesized that high viremia in crows leads to a high PBMC MOI relative to robins and, thus, higher polyinfection. We expected that this could facilitate the survival of rare and defective genomes via complementation in accordance with prior reports from studies conducted *in vitro* [33,52]. Similarly, we hypothesized that lower viremia in robins resulting in lower systemic MOI would lead to less PBMC polyinfection. Reduced polyinfection would then decrease the efficiency of complementation and reduce the survival of defective and rare genomes. To test these hypotheses, we assessed WNV replication in PBMCs and utilized a barcoded WNV to examine the replication of variably represented genotypes at varying MOIs.

WNV replicated to modest titers in PBMCs *ex vivo*, consistent with previous reports of equine and avian PBMC infections [39,40]. Despite higher viremia in crows, *ex vivo* infection of robin and crow PBMCs showed that robins generated similar to slightly higher levels of extracellular virus, suggesting that in robins, PBMCs might play a more significant role in virus replication, and other cell types or tissues may play a greater role in virus replication in crows *in vivo*. Crow PBMCs had the highest overall GE/PFU ratios during infection, indicating greater production of non-infectious particles. We also observed that higher MOIs in both robin and crow PBMC infections led to greater GE/PFU ratios. This is consistent with our previous *in vivo* studies that suggested high crow viremia may facilitate the persistence of defective genomes in the virus population [15]. Defective genomes are commonly generated in the lab and natural infections and have been demonstrated to survive multiple rounds of transmission through complementation by intact genomes, suggesting that their production in birds can lead to maintenance in nature [31,53,54,55].

We next generated barcoded WNV for use in evaluating MOI in vivo in birds to explore the unique landscape of virus variants in single cells. This barcoded WNV (BC-WNV), much like our previously published ZIKV barcoded virus [46], replicated similarly *in vitro* compared to wildtype WNV, indicating its suitability for *in vivo* studies. BC-WNV also exhibited *in vivo* phenotypes similar to wildtype, as demonstrated by crow and robin viremia and organ viral loads (not shown). When we performed NGS analysis of BC-WNV, we detected 2236 barcodes, significantly fewer than the theoretical maximum. The cause of this is unclear; however, inefficient transfection of 293T cells seems likely to have produced this result. Nevertheless, our data on BC-WNV replication *in vitro* and in birds, and our data on the complexity of the barcode region, indicated that it was a suitable tool to address the hypotheses we addressed in this work.

DF1 cells, crows and robins were infected with BC-WNV to determine the extent of polyinfection within cells of different origin. DF1 cells were infected at MOIs of 1 and 10 to compare *in vivo* findings to an established cell culture model of the avian–WNV interaction. DF1 cells contained up to 35,000 genome copies and had the highest mean genome copies per cell compared to avian PBMCs. This is not particularly surprising given that DF1 cell culture, while interferon-competent, is not affected by the influence of a full systemic innate immune response present during the *in vivo* infections performed to obtain PBMCs [42,56,57]. WNV genome copies and barcode counts within DF1 cells did not differ significantly between MOI 1 and 10, suggesting that DF1 cells provide suitable replication conditions that cause early saturation. Similarly, the generally higher titers in DF1 cells compared with PBMCs isolated from WNV-infected birds may be due to the absence of a complete innate immune response occurring in those cells *in vitro* or the fact that some of the isolated PBMCs may be refractory to WNV infection. Analysis of barcode frequencies within DF1 cells demonstrated that rare input barcodes were typically lost during replication, and that each DF1 cell was frequently dominated by a single barcode at a very high frequency, a trend not observed in crow- or robin-derived cells, indicating the limitations of DF1s as a model for WNV–avian interactions.

Analysis of viral loads within crow PBMCs compared to robin PBMCs revealed striking differences in both genome copy numbers and total barcode counts. Crow PBMCs had higher genome copy numbers while robins contained the least, with an approximately 100-fold mean difference in viral load in cells from each species. A single crow PBMC is capable of containing up to 388,000 genome copies, while robin PBMCs contained up to 2,000 genome copies. This high viral load in crow PBMCs compared to robin PBMCs is consistent with viremia patterns in these animals and supports the role of PBMCs in generating host-specific viremia and mortality phenotypes in host animals. Moreover, these data indicate that clinically susceptible animals that develop strikingly high viremias contain high viral loads in circulating PBMCs.

Further, crows had higher numbers of unique barcodes within individual cells, indicating more frequent polyinfection compared to robins. Comparison of changes in barcode frequency in the input inoculum to crow PBMCs revealed that rare barcodes in the input virus may rise in frequency in crow cells, sometimes to greater than 50% of the cell-specific population. In contrast, rare mutations in the initial stock tended to not be detected in robin PBMCs. We also observed more unique barcodes and higher levels of barcode diversity and complexity in crow compared to robin PBMCs. These data permit us to conclude that in crows, frequent polyinfection of PMBCs, and likely other cell types, facilitates the persistence of genetic diversity in these animals, including defective viral variants that may reduce the average fitness of the virus population. Moreover, hosts that experience high viremias may be key to maintaining virus genetic diversity at the population level and decrease the strength of purifying selection. As a consequence, while low-fitness variants may be maintained, high-fitness variants may also be prevented from rising in frequency within the population. Conversely, infection of robins results in less frequent polyinfection and an overall reduction in population variation that may either result in fitness increases due to the action of purifying selection or bottlenecks that lead to fitness declines over time. These somewhat divergent selective environments in crows and robins lead us to hypothesize that WNV variants of reduced fitness would persist longer within crows compared to robins, a prediction that is supported by our prior work [15]. Testing this hypothesis directly requires additional study.

In summary, this study quantifies and characterizes the extent that ‘host susceptibility,’ which here refers to viremia level and mortality, is associated with virus evolutionary dynamics in ecologically relevant hosts. Our work with PBMCs demonstrated that individual cells, particularly in crows, can contain extremely high viral load and virus diversity, providing an environment that permits the persistence of defective genomes, likely via complementation.

## Figures and Tables

**Figure 1 pathogens-12-00767-f001:**
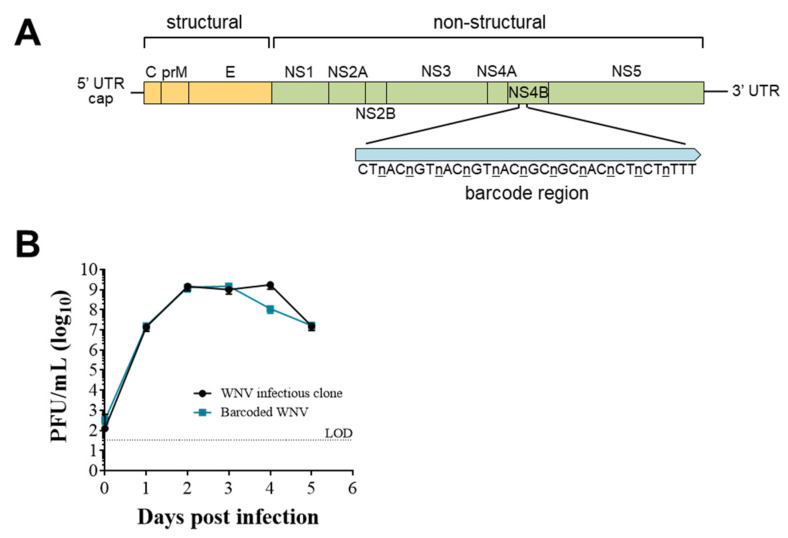
**Establishing and characterizing WNV barcoded virus (BC-WNV).** (**A**) Schematic diagram of BC-WNV genome depicting the location and content of the barcode region. The NS4B region of the genome was engineered to contain a region (7237–7269 bp) with 11 consecutive synonymous degenerate nucleotides at every third codon position. (**B**) Growth curve of viruses derived from the parental WNV infectious clone and BC-WNV in Vero cells (MOI = 0.1).

**Figure 2 pathogens-12-00767-f002:**
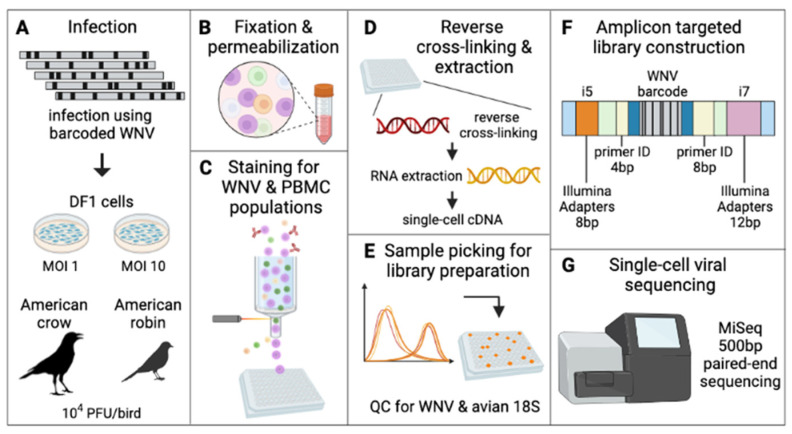
**Approach to barcode quantification of infected DF1 cells and avian PBMCs via flow cytometry and amplicon targeted library construction.** (**A**) DF1 cells at MOI 1 and 10 and American crows and robins at 10,000 PFU/bird were infected with BC-WNV. (**B**) PBMCs collected from whole avian blood, along with DF1 cells, were separated, fixed and permeabilized. (**C**) Cells were stained for WNV viral protein and cell type and sorted into a 96-well plate. (**D**) Cells were reverse cross-linked and viral RNA was extracted and reverse-transcribed into cDNA. (**E**) Individual cells were classified as WNV infected by qRT-PCR and avian 18S. (**F**) Libraries were constructed by adapting the Primer ID approach to the Illumina MiSeq platform (Appendix A). (**G**) Sequencing was performed on an Illumina platform.

**Figure 3 pathogens-12-00767-f003:**
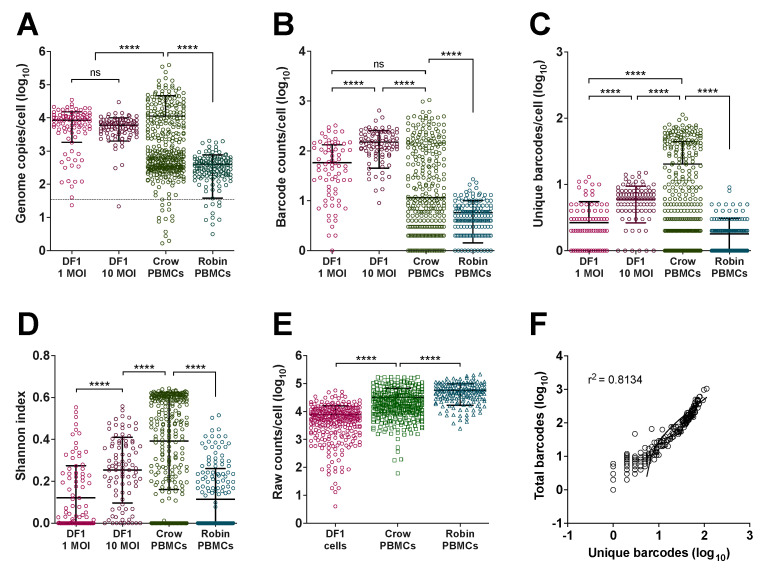
**Viral Barcode RNA Abundance and Complexity Varies Between Cells and Across Wild Bird Hosts.** (**A**) WNV Genome copies per cell (log10) in BC-WNV-infected DF1 cells at MOI 1 and 10 and crow and robin PBMCs, as determined by qRT-PCR for E gene. (**B**) Total barcode counts per cell (log10) in BC-WNV-infected DF1 cells at MOI 1 and 10 and crow and robin PBMCs, as determined by barcode sequencing. (**C**) Number of unique barcodes per cell (log10) in BC-WNV-infected DF1 cells at MOI 1 and 10 and crow and robin PBMCs, as determined by barcode sequencing. (**D**) Shannon index (complexity) in DF1 cells at MOI 1 and 10 and crow and robin PBMCs. (E) Sequencing depth per cell (log10) in DF1 cells, crow and robin PBMCs. ((**A**–**E**), **** *p* < 0.0001, ANOVA) (**F**) Correlation of unique to total barcodes (log10) (r^2^ = 0.8134).

**Figure 4 pathogens-12-00767-f004:**
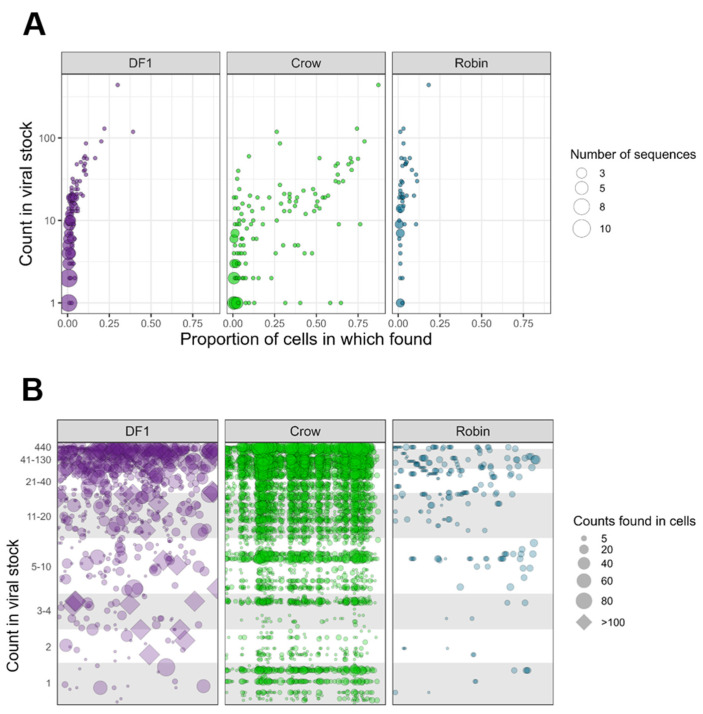
**Barcode Prevalence Varies Between Cells and Across Wild Bird Hosts.** (**A**) Frequency of each unique barcode in BC-WNV inoculum compared to detection in cells after replication. (**B**) Counts of unique barcode sequences in BC-WNV inoculum compared to counts found in cells after replication.

## Data Availability

The data generated as part of this project are available from the authors upon request and will be deposited in the NCBI SRA upon manuscript acceptance.

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
