# Peer review of "Intracellular Diversity of WNV within Circulating Avian Peripheral Blood Mononuclear Cells Reveals Host-Dependent Patterns of Polyinfection"

_pathogens, 2023, doi:10.3390/pathogens12060767_

Round 1

Reviewer 1 Report

The manuscript by Frank et al. entitled “Intracellular diversity of WNV within circulating avian peripheral blood mononuclear cells reveals host-dependent patterns 4 of polyinfection” is a well written manuscript with interesting results that logically follow recently previous work from the Ebel group. There are several suggestions below that would improve the quality of the manuscript.

Comments: 

1)     The Introduction provides a useful overview WNV infections in crows and robins, but there is no information regarding how this study relates to human disease, a major focus of Pathogens. The authors should provide some information regarding the relevance of this study in the context of human disease.

2)     The Introduction would benefit from very brief inclusion of the WNV tropism, especially cell types within PBMC. Along these lines, it would also be informative to include in the Results flow cytometry scatter plots with indication of anti-WNV+ cells as well as histograms showing the number of cells anti-WNV+ (assuming no cell-specific markers were assessed by flow cytometry).

3)     It is unclear how FtC-3699 was propagated (cell types, passage number, etc). This information should be included.

4)     For the screening of WNV positive cells, it is not clear if all cells, regardless of Ct value, are included for sequencing. This should be clarified.

5)     In the supplement, there is a legend titled “Supplemental Methods Figure 1” but there are no associated tables. These tables need to be included.

6)     There is no indication regarding the bioinformatics pipeline(s) used to analyze the sequencing results.

Author Response

The Introduction provides a useful overview WNV infections in crows and robins, but there is no information regarding how this study relates to human disease, a major focus of Pathogens. The authors should provide some information regarding the relevance of this study in the context of human disease.

We have added text to the introduction that provides some context to how our studies are related to human health (Lines 42-45).

The Introduction would benefit from very brief inclusion of the WNV tropism, especially cell types within PBMC. Along these lines, it would also be informative to include in the Results flow cytometry scatter plots with indication of anti-WNV+ cells as well as histograms showing the number of cells anti-WNV+ (assuming no cell-specific markers were assessed by flow cytometry).

We have added additional information in the introduction regarding WNV tropism and cell types within avian PBMCs (lines 78-80).

We have included an additional supplemental figure (Supplemental Figure 2) showing flow histograms of DF1 cells that were uninfected or infected with WNV an MOI of 1 and 10. This figure is discussed on line 279.

It is unclear how FtC-3699 was propagated (cell types, passage number, etc). This information should be included.

We have added additional information on this virus on lines 92-94.

For the screening of WNV positive cells, it is not clear if all cells, regardless of Ct value, are included for sequencing. This should be clarified.

We have clarified in the methods that cells with any detectable WNV viral RNA as determined by qRT-PCR, regardless of cycle threshold value, were included for sequencing (line 177). We added additional discussion on lines 278-280, and 290-292.

In the supplement, there is a legend titled “Supplemental Methods Figure 1” but there are no associated tables. These tables need to be included.

We thank the reviewer for finding this omission. We have added these tables to the Supplemental Methods.

There is no indication regarding the bioinformatics pipeline(s) used to analyze the sequencing results.

We thank the reviewer for finding this omission. We have added a bioinformatics pipeline to the methods section (lines 219-231).

Reviewer 2 Report

In this manuscript, Frank & Byas et al. assessed WNV infection dynamics and genomic diversity in WNV-infected PBMCs derived from robins and crows to gain insights on virus evolution in hosts who exhibit different levels susceptibility. Overall, I applaud the authors on their work and feel their conclusions are justified. Some minor thoughts/clarifications are listed below:

- As the authors emphasize, robins exhibit significantly lower WNV viremia than in crows, yet in Figure S1, I was surprised that WNV viral loads were similar or sometimes even higher for robin PBMCs compared to crow PBMCs in vitro (lines 225-226). Could the authors comment why this discrepancy may be, given that PBMCs are "key targets of WNV replication in mammals and birds" (line 76)? Also, please carefully cite Figure S1, including the appropriate A-I, when the results are reported in lines 223-233. I felt this section was confusing.

- Figure 1B: In addition to looking at Vero cells, is it possible to compare or comment on the growth curves of the barcoded WNV/the WNV infectious clone and the FtC-3699 WNV strain in the DF1 or bird PBMCs (similar to Supplementary Figure 1), at least at the MOI/time point shown in Figure 2?

- Figure 3C: Those cells that have zero unique barcodes, these are uninfected cells, correct? I question whether uninfected cells should be included in the statistical analysis for barcode diversity/complexity. Could those populations be excluded?

Author Response

  1. As the authors emphasize, robins exhibit significantly lower WNV viremia than in crows, yet in Figure S1, I was surprised that WNV viral loads were similar or sometimes even higher for robin PBMCs compared to crow PBMCs in vitro (lines 225-226). Could the authors comment why this discrepancy may be, given that PBMCs are "key targets of WNV replication in mammals and birds" (line 76)? Also, please carefully cite Figure S1, including the appropriate A-I, when the results are reported in lines 223-233. I felt this section was confusing.

We have commented on the discrepancy between robin and crow PBMC in vitro replication versus what is seen in birds on lines 363-367 of the discussion, noting that perhaps in robins, other cell types and tissues might contribute more to viremia than in crows. We have more thoroughly cited panels in Figure S1 (lines 241-249).

  1. Figure 1B: In addition to looking at Vero cells, is it possible to compare or comment on the growth curves of the barcoded WNV/the WNV infectious clone and the FtC-3699 WNV strain in the DF1 or bird PBMCs (similar to Supplementary Figure 1), at least at the MOI/time point shown in Figure 2?

We did not do any experiments directly comparing growth curves with the barcoded/infectious clone WNV, and FtC-3699 WNV strain in DF1 cells or PBMCs (ex vivo nor in vivo). However, based on unpublished data from the lab, expect the barcoded WNV to replicate similarly in those cells as the WNV isolate.

In the experiment outlined in Figure 2, we only collected cells for intracellular viral RNA, and did not examine levels of extracellular virus (as was done in Supplementary Figure 1) so we unfortunately cannot comment on the replication of virus in DF1 cells, American robins and American crows.

We have noted on line 239 that the experiments in Supplemental Figure 1 use the WNV isolate FtC-3699 strain.

  1. Figure 3C: Those cells that have zero unique barcodes, these are uninfected cells, correct? I question whether uninfected cells should be included in the statistical analysis for barcode diversity/complexity. Could those populations be excluded?

We have added additional discussion clarifying that all cells analyzed were positive for WNV viral protein via flow cytometry and that while all cells were also positive for WNV viral RNA via qRT-PCR (Figure 3A), some cells did not have any detectable barcodes (lines 278-280, 290-292).

We chose to include those cells because they not only make up a very small percentage of the total cells/group, but they were virus positive using the other two assays (flow cytometry and qRT-PCR).

Round 2

Reviewer 1 Report

No additional comments.